



# Case Studies of the Impact of Orbital Sampling on Stratospheric Trend Detection and Derivation of Tropical Vertical Velocities: Solar Occultation versus Limb Emission Sounding

Luis F. Millán[1], Nathaniel J. Livesey[1], Michelle L. Santee[1], Jessica L. Neu[1], Gloria L. Manney[2,3], and Ryan A. Fuller[1]

[1]Jet Propulsion Laboratory, California Institute of Technology, Pasadena, California, USA
[2]New Mexico Institute of Mining and Technology, USA
[3]NorthWest Research Associates, USA

*Correspondence to:* L. Millán (luis.f.millan@jpl.nasa.gov)

**Abstract.** This study investigates the representativeness of two types of orbital sampling applied to stratospheric temperature and trace gas fields. Model fields are sampled using real sampling patterns from the Aura Microwave Limb Sounder (MLS), the HALogen Occultation Experiment (HALOE) and the Atmospheric Chemistry Experiment Fourier Transform Spectrometer (ACE-FTS). The MLS sampling acts as a proxy for a dense uniform sampling pattern typical of limb emission sounders, while HALOE and ACE-FTS represent coarse non-uniform sampling patterns characteristic of solar occultation instruments. First, this study revisits the impact of sampling patterns in terms of the sampling bias, as previous studies have done. Then, it quantifies the impact of different sampling patterns on the estimation of trends and their associated detectability. In general, we find that coarse non-uniform sampling patterns may introduce non-negligible errors in the inferred magnitude of temperature and trace gas trends and necessitate considerably longer records for their definitive detection. Lastly, we explore the impact of these sampling patterns on tropical vertical velocities derived from stratospheric water vapor measurements. We find that coarse non-uniform sampling may lead to a biased depiction of the tropical vertical velocities and, hence, to a biased estimation of the impact of the mechanisms that modulate these velocities. These case studies suggest that dense uniform sampling such as that available from limb emission sounders provides much greater fidelity in detecting signals of stratospheric change (for example, fingerprints of greenhouse gas warming and stratospheric ozone recovery) than coarse non-uniform sampling such as that of solar occultation instruments.

## 1 Introduction

Satellite data have provided a wealth of information on the Earth system and have had a profound impact on operational numerical weather forecasting. Unlike ground-based instruments or airborne field campaigns, satellite data provide continuous global coverage, which facilitates the study and assimilation of distributions of atmospheric fields, as well as global model evaluation. However, satellite measurements sample continuously changing atmospheric fields only at discrete times and lo-





cations, depending on the satellite orbit as well as the measurement technique, which can result in a biased depiction of the atmospheric field.

Typically, the impact of orbital sampling has been evaluated by comparing a raw model field against a satellite-sampled one. For example, many studies have documented sampling errors for rainfall estimates (e.g., McConnell and North, 1987; North

et al., 1993; Bell and Kundu, 1995; Soman et al., 1996; Gebremichael and Krajewski, 2005), and brightness temperatures (Engelen et al., 2000; Brindley and Harries, 2003), as well as $O_3$, CO, temperature and a few other atmospheric parameters sampled by nadir-viewing instruments (Luo et al., 2002; Aghedo et al., 2011; Guan et al., 2013). Recently, Toohey et al. (2013) evaluated the sampling bias in monthly and annual mean climatologies of $O_3$ and $H_2O$ from 16 satellite instruments, including limb emission sounders, limb scattering sounders, solar occultation instruments and a stellar occultation instrument.

They concluded that coarse sampling may introduce significant sampling uncertainties in climatologies, not only through non-uniform spatial sampling but, more importantly, through non-uniform temporal sampling, that is to say, producing regional monthly means using measurements that do not cover the entire month. As expected, the sampling bias was found to be the greatest in regions with large natural variability.

In this study we evaluate the impact of the Aura Microwave Limb Sounder (MLS), the HALogen Occultation Experiment

(HALOE) and the Atmospheric Chemistry Experiment Fourier Transform Spectrometer (ACE-FTS) sampling patterns using the Canadian Middle Atmosphere Model (CMAM). MLS sampling provides a dense uniform pattern, while HALOE and ACE-FTS are representative of coarser solar occultation sampling patterns. We use HALOE and ACE-FTS sampling patterns because they are commonly used solar occultation datasets and, furthermore, because their sampling patterns are significantly different, and thus representative of the range of observation patterns obtained by solar occultation instruments.

Our study has two purposes: (1) We expand upon previous studies by quantifying the sampling bias of these instruments affecting measurements of upper tropospheric and stratospheric temperature and trace gas species. (2) We investigate how differences in data coverage may affect the outcome of two illustrative atmospheric studies: trend detection and quantification of tropical vertical velocities. We assess the differences in the long-term (>30 years) trends in temperature, $O_3$, and CO estimated using datasets with different sampling patterns. We find that sampling-induced spurious features can affect the accuracy of

these trends, which in turn affects estimates of their magnitude as well as their detectability. Also, we characterize the impact of orbital sampling on derived lower stratospheric tropical vertical velocities. These velocities are computed by correlating the lag of the water vapor "tape recorder" signal between adjacent levels (Niwano et al., 2003; Flury et al., 2012; Jiang et al., 2015). As such, they are likely an upper bound on the actual velocity (Schoeberl et al., 2008). These vertical velocities are modulated by the Quasi-Biennial Oscillation (QBO), seasonal cycles, and El Niño Southern Oscillation (ENSO) (e.g., Flury

et al., 2013; Neu et al., 2014; Minschwaner et al., 2016). We show that sampling-induced spurious features affect the vertical velocity estimates, which in turn will affect estimates of the magnitude of variability in the circulation associated with those oscillations.

This paper is organizes as follows: Section 2 describes the satellite patterns and the model fields used. Section 3 briefly revisits sampling bias estimates, while the impact of sampling on long-term trends as well as in trend detection is presented in

section 4. Section 5 addresses the impact of orbital sampling on derived tropical vertical velocities, and section 6 summarizes





our results. The results discussed in this study should be considered as example cases. Whether the results shown represent reasonable estimates of the true orbital sampling induced artifacts (e.g. in the sampling bias, in the inferred magnitude of the trends, or in the derived tropical vertical velocities) may also depend on how well the model fields represent the real atmosphere.

## 2 Data and Methodology

### 2.1 Model Fields

CMAM is used as a proxy for the real atmosphere. CMAM is an extension of the Canadian Center for Climate Modeling and Analysis spectral general circulation model. Detailed descriptions of its dynamical and chemical schemes are given by Beagley et al. (1997) and de Grandpré et al. (2000), respectively. The free-running version of the model has been extensively evaluated, and has been shown to agree relatively well with observations relevant to chemistry, dynamics, transport, and radiation (e.g., de Grandpré et al., 2000; Eyring et al., 2006; Jin et al., 2005; Hegglin and Shepherd, 2007; Melo et al., 2008; Jin et al., 2009).

In this study we use output from the CMAM30 Specified Dynamics (SD) simulation in which temperature and winds have been nudged to the ERA-Interim reanalysis. This data set exploits the vast progress made by reanalyses in representing the stratospheric circulation (e.g., Dee et al., 2011) and as such can be used to reliably predict the chemical fields. Before nudging the temperature fields, a technique described by McLandress et al. (2014) was used to remove temporal discontinuities in the ERA-Interim upper stratospheric temperatures that occurred in 1985 and 1998. CMAM30-SD has been shown to have a good representation of stratospheric temperature, $O_3$, $H_2O$ and $CH_4$ (Pendlebury et al., 2015), it has been used as a transfer function between satellite datasets to construct a reliable long-term $H_2O$ data record (Hegglin et al., 2014), and it has been shown to reproduce halogen-induced $O_3$ loss sufficiently well for investigation of long-term $H_2O$ trends (Shepherd et al., 2014). The version of CMAM30-SD used here has a horizontal resolution of approximately 3.75° latitude by 3.75° longitude. This resolution (approximately 400 km) is comparable to the spatial resolution of HALOE, ACE-FTS and MLS, which is limited by the ∼500 km limb-viewing path length, and, hence, no smoothing of the model fields is necessary (Toohey et al., 2013). This version has 63 vertical levels up to 0.0007 hPa with a vertical resolution varying from 100 m in the lower troposphere to about 3 km in the mesosphere. Model results for the period between January 1979 and December 2012 are used in this study.

We evaluate the following CMAM30-SD outputs: Temperature, $O_3$, $CH_3Cl$, $H_2O$, CO, HCl, $N_2O$, and $HNO_3$. These parameters are an intersection of the available CMAM30-SD outputs, the measurements available for MLS, and the measurements available for ACE-FTS or HALOE.

### 2.2 Satellite Instrument Sampling Patterns

In this study we analyze the representativeness of the orbital sampling of the solar occultation instruments HALOE and ACE-FTS, as well as the limb emission sounder MLS. Solar occultation data are extremely valuable for atmospheric studies due to their fine vertical resolution, the excellent precision and accuracy of their self-calibrated measurements, and their potential for detecting many species. However, the sparsity of the measurements makes understanding the impact of their sampling crucial.



HALOE was launched on the Upper Atmosphere Research Satellite (UARS) in 1991, and it measured infrared spectra across eight broadband and gas filter channels from 2.45 $\mu$m to 10.04 $\mu$m for 14 years. It measured vertical profiles of temperature, pressure and several atmospheric trace gases, with as many as 15 sunrise and 15 sunset profiles of these atmospheric parameters observed at a given latitude each day (Russell et al., 1993). The HALOE sampling sweeps through its full range of latitude coverage, ranging from $\pm 80^o$ to $\pm 50^o$ depending on the season, over a period of about a month.

ACE-FTS was launched in 2003 and profiles the atmosphere by using solar occultation. It measures infrared spectra from 2.2 to 13.3 $\mu$m (750 to 4400 cm$^{-1}$) with high spectral sampling (0.02 cm$^{-1}$), which allows retrieval of temperature, pressure and concentration for several dozen atmospheric trace gases (Bernath et al., 2005). ACE-FTS is focused on high-latitude science, and thus almost 50% of its approximately 15 sunrise and 15 sunset occultations per day occur at latitudes around 60$^o$. Global latitude coverage is achieved over a period of approximately three months.

Aura MLS was launched in 2004 and measures limb millimeter and submillimeter atmospheric thermal emission using heterodyne radiometers covering spectral regions near 118, 191, 240, 640 GHz and 2.5 THz, from which temperature, trace gas concentrations and cloud ice are retrieved. Daily, it covers latitudes from 82$^o$S to 82$^o$N with $\sim$3500 vertical scans providing near-global observations.

To investigate the impact of orbital sampling, the daily model fields are linearly interpolated to the actual latitude and longitude of the satellite measurements. For the sampling patterns, we use a typical year of measurement locations. In particular, we use 1994, 2005 and 2008 for HALOE, ACE-FTS and MLS, respectively; these are the years with the maximum number of measurements on record for each dataset. Gaps in the measurements due to instrument problems as well as year-to-year variations due to orbital state changes are not considered in this study. To avoid differences attributed purely to diurnal cycles, all satellite measurements are assumed to be made at 12UT, obviating the need for interpolation in time. Thus, we focus only on spatial differences. Given that our focus is in horizontal/temporal sampling, all satellite measurements are assumed to have the same vertical resolution as CMAM30-SD. That is, the impact of the averaging kernels is not addressed in this study.

Figure 1 (left) shows monthly sampling counts for each instrument. MLS has a dense and nearly uniform sample density over latitude and time, while HALOE and ACE-FTS have sparser and less uniform sample densities because they are limited to two measurements per orbit. Figure 1 (right) shows the zonal mean water vapor field at 100 hPa as sampled by each instrument to highlight how much daily variability may be missed by the HALOE and ACE-FTS sampling patterns. The consequences of these contrasting sampling densities are the main motivation for this study. As discussed by Manney et al. (2007), mapping data into vortex-centered coordinate systems such as those based on potential vorticity (PV) or equivalent latitude (EqL) may alleviate some of the solar occultation sampling density problems for polar processing studies. However, since this study focuses on near-global trends and tropical upwelling velocities, such vortex-centered coordinate systems are of very limited utility here.





## 3 Sampling Biases

We evaluate the sampling biases associated with constructing monthly zonal means from the raw and satellite-sampled data. The raw or sampled zonal means for a particular latitude bin for each pressure level are given by

$$\overline{Z_l^x} = \frac{1}{N} \sum y_l^x \qquad (1)$$

where $N$ is the total number of points belonging to a latitude bin $l$, and x is either the raw data, denoted by the superscript $r$, or the sampled data, denoted by the superscript $s$. Figure 2 shows examples of raw and sampled zonal means for Temperature, $O_3$ and $H_2O$. The difference between the satellite-sampled zonal mean and the raw zonal mean gives the absolute sampling bias, that is to say,

$$S_A = \overline{Z_l^s} - \overline{Z_l^r} \qquad (2)$$

or, in percentage,

$$S_P = \frac{\overline{Z_l^s} - \overline{Z_l^r}}{\overline{Z_l^r}} \times 100. \qquad (3)$$

Figure 3 shows examples of the sampling biases for Temperature, $O_3$ and $H_2O$ for January 2005. Relative biases are shown for trace gas species to accommodate their strong vertical gradients. These biases only display the impact of sampling the CMAM30-SD fields; as mentioned before, how well these biases represent the true atmospheric sampling biases will depend on how close the model fields are to the real atmospheric state.

For each month, instrument, and pressure level, this bias was computed for all the latitude bins in which an instrument was able to sound the atmosphere. To summarize the potential sampling biases we computed root-mean-square (RMS) biases over one year's worth of data. As an example, Figure 4 shows these calculated RMS sampling biases for Temperature, $O_3$ and $H_2O$ for 2005. Overall, there is a direct correlation between the sampling biases and the variability of the geophysical parameters. For example, as noted by Toohey et al. (2013), $O_3$ sampling biases for the three instruments are smaller in the tropics and larger at midlatitudes and in the polar regions, where variability is low/high, respectively. However, the biases in all regions are minimized by dense uniform sampling such as that of MLS.

Figure 5 shows the mean and maximum sampling biases over all latitudes for the year 2005 for all the atmospheric parameters studied. In general, HALOE and ACE-FTS sampling patterns produce mean and maximum sampling biases an order of magnitude larger than those of MLS. For example, for the occultation sensors, the Temperature maximum sampling biases are about $10\,K$ compared to $1\,K$ for MLS. Similarly, in the middle stratosphere, $H_2O$ maximum sampling biases for the solar occultation instruments can be as large as $5\,\%$ compared to less than $1\,\%$, and lastly, $HNO_3$ maximum sampling biases can be as large as $50\,\%$ compared to less than $5\,\%$.

## 4 Long-Term Trends

We now evaluate the impact of orbital sampling on the representation of long-term trends. Accurate representation of long-term trends is crucial because they are indicators of climate change, as well as ozone recovery. To summarize the effect of the orbital



sampling upon long-term trends we use Taylor diagrams (Taylor, 2001), which provide a convenient method for visualizing statistics of how closely patterns match each other; in this case, they are used to depict the success of the satellite-sampled data in representing the variability found in the raw model fields. The similarity is quantified by their correlation coefficient, their centered RMS difference (RMSd) and their standard deviations.

In the diagrams shown, everything is normalized to the raw-model standard deviation to facilitate showing different pressure levels in the same figure. In these diagrams, there are four things to consider: (1) the azimuth angle indicates the correlation between the satellite-sampled and raw data, (2) the point with normalized standard deviation of one and correlation of one is the reference point and corresponds to the raw model data, (3) the distance between any point in the figure and the reference point indicates the ratio of the centered RMSd and the raw-model standard deviation (green contours), and (4) the distance

between other points in the plot and the origin is the ratio between that satellite-sampled standard deviation and that of the raw model field.

Near-global (60°S-60°N) long-term (1979–2012) trends are compared between satellite-sampled and raw model fields in Figure 6 for all the atmospheric parameters evaluated in this study. We did not expand this study to the latitudes poleward of 60°N or 60°S because ACE-FTS does not sample these areas for four months per calendar year and HALOE does not

sample for 5 and 6 months at the South and North Pole, respectively (see Figure 1). Figure 7 shows the the raw model standard deviations used to normalize these diagrams (black lines). Overall, the MLS-sampled data (circles in Figure 6) for all variables and all pressure levels are close to the reference point, indicating high correlation coefficients, low centered RMS differences and the expected standard deviation (i.e., a standard deviation similar to that of the full model fields). The HALOE-sampled data (triangles) show intermediate performance, followed by the ACE-FTS-sampled data (squares), which show the weakest

correlation and the largest normalized standard deviation. For example, this is easily seen in the CO Taylor diagram, where the MLS-sampled points all cluster tightly at the reference point, whereas HALOE-sampled points lie farther away and ACE-FTS-sampled points the farthest.

To highlight the impact of these sampling differences, Figure 8 shows trend estimates for near-global temperature at 10 hPa using the raw and satellite-sampled data. Three methods have been used to compute the trends. The first is a simple linear fit

through the points. In the second, we deseasonalize the data (we remove the observed climatological monthly mean at every grid point) before computing a linear fit. Lastly, we consider a model of the form,

$$Y = \mu + \omega \frac{t}{12} + S + N \tag{4}$$

where $Y$ are the monthly raw or sampled average measurements (Temperature, CO or $O_3$ concentration, etc), $\mu$ is a baseline constant, $\omega$ is the mean trend per year, $t$ is time in months, $S$ is a seasonal mean component represented by

$$S = a_1 sin(\frac{2\pi}{12} t + b_1) + a_2 sin(\frac{2\pi}{6} t + b_2) \tag{5}$$

and $N$ is the unexplained portion of the data assumed to follow a first order autoregressive model [AR(1)]. That is, it satisfies

$$N = \phi N_1 + \varepsilon \tag{6}$$





where $\phi$ is the autocorrelation of the noise and $\varepsilon$ are independent white noise variables with variance $\sigma_\varepsilon^2$. As pointed out by Tiao et al. (1990), $\phi$ has the effect of reducing the amount of information that would have been available in the same number of independent data points. Similar models have been used in many previous trend studies (Tiao et al., 1990; Weatherhead et al., 1998; Boers and Meijgaard, 2009; Whiteman et al., 2011). As shown in Figure 8, HALOE (ACE-FTS) sampling artificially

reduces (increases) the trend estimates by about 10% (25%). Despite agreement on the sign of the trend, these sampling-induced artifacts will compromise the robustness of the derived temperature trends. We computed the trend using different methods in order to emphasize that using models that are more geophysically realistic, such as those that capture the seasonal component, may not have much impact on the estimated trends, or, as pointed out by Weatherhead et al. (1998), on the trend statistical properties.

Figure 9 shows how these sampling-induced trend artifacts vary with altitude. To avoid clutter, this figure only shows the differences in trend magnitude computed using equation 4, but the ones computed using the other trend detection methods are similar. We show results for Temperature, $O_3$ and CO because these parameters exhibit clear trends at most pressure levels in the CMAM30-SD simulations and also because overall they can be accurately described by the model given by equation 4. For $O_3$ we only use data starting from 2000 to capture the expected period of $O_3$ recovery. Overall, MLS sampling allows

estimation of the trend magnitudes to about an order of magnitude better than HALOE and ACE-FTS sampling, with accuracy better than 1% at most pressure levels for temperature and CO, and better than 10% for $O_3$.

Figure 9 also shows the estimated number of years required to definitively detect these trends. When using equation 4, the number of years, $n^*$, needed to detect a given trend with a 95% confidence level with probability of 0.90 can be approximated by (Tiao et al., 1990),

$$20 \quad n^* = \left[ \frac{3.3\sigma_N}{|\omega|} \sqrt{\frac{1+\phi}{1-\phi}} \right]^{2/3} \tag{7}$$

which indicates that trend detectability depends on three factors: (1) $\phi$, the autocorrelation of the noise, (2) $\sigma_N$, the standard deviation of the total noise in the time series, which corresponds to the unexplained variability of the data, and (3) the absolute magnitude of the trend. It is also noted that $\sigma_N$ is related to $\sigma_\varepsilon$ by

$$\sigma_N^2 = \frac{\sigma_\varepsilon^2}{1-\phi^2} \tag{8}$$

in this model. As shown in Figure 9, trend detection using data with HALOE or ACE-FTS sampling will require considerably more years than using data with MLS sampling. This is due to an increase in the magnitude of $\sigma_N$ resulting from the noisiness of the time series based on the HALOE or ACE-FTS sampling patterns (e.g. Figure 8). For example, at 1 hPa, the pressure level where the strongest temperature trend is found in CMAM30-SD, a 15-year record of MLS-sampled observations would be required to detect such a trend at the 95% confidence level, while HALOE and ACE-FTS sampling would require 25 and

30 years, respectively. For $O_3$ at 2 hPa, the pressure level where the strongest $O_3$ trend is found in CMAM30-SD, the MLS sampling pattern would require about 11 years, while HALOE and ACE-FTS would require about 20 and 30 years, respectively. In addition, MLS sampling requires the same number of years as for the raw model fields; that is, the required number of years is only determined by the natural variability.



We also investigated the effect of instrument noise, using Tiao et al. (1990) and Whiteman et al. (2011)

$$\sigma_N = \sqrt{\frac{\sigma_\varepsilon^2}{1-\phi^2} + \frac{\sigma_I^2}{n_I}} \tag{9}$$

where $\sigma_I$ is the instrument noise and $n_I$ is the number of measurements averaged. Typical noise estimates were taken from Livesey et al. (2015) for MLS; Clerbaux et al. (2005), Sica et al. (2008), and Dupuy et al. (2009) for ACE-FTS; and Hervig
et al. (1996) and Brühl et al. (1996) for HALOE. Since HALOE does not measure CO, we assumed the same error as given by Clerbaux et al. (2005) for ACE-FTS. The effect of instrument noise was found to be negligible due to the high number of measurements even for HALOE and ACE-FTS (in a given month, around 70000 for MLS, 600 for HALOE and 270 for ACE-FTS). These estimates of the length of the measurement record required to detect trends do not take into account the effects of a disruption of the measurements for a given period or aging of the instrument, both of which can induce artificial trends in
the data that are not representative of the actual environmental trend studied. In addition, data gaps associated with clouds will exacerbate the solar occultation sampling problems in the upper troposphere, whereas microwave emission measurements can penetrate through aerosol and all but the thickest clouds.

Both HALOE and ACE-FTS provide better coverage in the extratropics than in the tropics (see Figure 1). Figure 10 therefore shows long-term trend comparisons between satellite-sampled and raw data for trends derived using only data from 30° N to
60° N. Figure 7 also shows the raw model standard deviations used to normalize these diagrams (purple lines). In general, HALOE and ACE-FTS sampled data correlation coefficients improved considerably over the near-global case, with correlation coefficient no smaller than ∼0.6 and with centered RMSd better than one raw model standard deviation (see Figure 10). MLS-sampled data are still closest to the reference point.

Figure 11 is equivalent to Figure 9 but for the 30° N to 60° N latitude range. As for the near-global trends, ACE-FTS sampling
still requires considerably more years to confidently detect a trend than does MLS sampling. HALOE, however, has a more uniform sampling density than ACE-FTS in this latitude range (see Figure 1), and thus the time required to detect a trend is more in line with that for MLS. Nevertheless, MLS sampling allows estimation of trends to about an order of magnitude better than HALOE and ACE-FTS sampling. As before, the effect of instrument noise was found to be negligible (for this latitude range the approximate number of measurements in a given month is 19000, 220 and 70 for MLS, HALOE and ACE-FTS,
respectively).

## 5  Tropical Vertical Velocities

In this section we investigate the impact of orbital sampling upon derived tropical vertical velocities (a key metric for atmospheric circulation). The vertical velocities are calculated using the same approach as described by Flury et al. (2012) and Jiang et al. (2015). In short, we use time series of daily zonal mean water vapor data averaged between 8°S and 8°N (see
Figure 12). We correlate these time series at different pressure levels and determine the time lag for the best correlation. The vertical velocity for the midpoint of each layer is simply computed by dividing the distance between the pressure levels (the altitude difference) by the lag. These calculations were performed using the raw model CMAM30-SD simulations as well as





the satellite-sampled data. Interpolation was used to fill the data gaps due to the sampling patterns. In the case of HALOE sampling, this implies linearly interpolating to fill gaps in June and December, while for ACE-FTS gaps are filled in January, March, May, July, September, November and December, when no measurements are made over the tropics (8°S to 8°N). The vertical velocities derived from this method are a measure of the transport velocity averaged over 8°S-8°N and have been

shown to agree well with the Transformed Eulerian Mean residual vertical velocity when in-mixing from the extratropics and vertical diffusion are small (Schoeberl et al., 2008).

Figure 13 (top) shows the vertical velocities averaged over 60-30 hPa derived using raw model fields as well as the satellite-sampled data. To quantify the impact of the different orbital sampling patterns, Figure 13 (bottom) displays scatter plots between the raw fields and the satellite-sampled vertical velocities. The best correlation (R = 1.00), the best line fit (0.93x+0.02)

and the smallest RMSd (0.005) are found when using the MLS sampling. ACE-FTS and HALOE sampling lead to non-negligible artifacts when deriving vertical velocities from the tape recorder.

Previous studies have shown variability in middle stratospheric tropical vertical velocities on the order of up to $\pm$ 40% associated with the QBO and ENSO (Flury et al., 2013; Neu et al., 2014; Minschwaner et al., 2016). To better understand the impact of these sampling-induced artifacts, we fit the following model to the monthly vertical velocities

$$w_{TR} = q * QSI[t - t_q] + e * MEI[t - t_e] + c \tag{10}$$

where $w_{TR}$ is the vertical velocity derived from the tape recorder, QSI is a QBO shear index, MEI is the multivariate ENSO index, $c$ is a baseline constant, $q$ and $e$ are constants modifying the magnitude of the QSI or MEI, and, $t_q$ and $t_e$ are the QSI or MEI time offsets, respectively. The QSI is calculated from the difference in the Singapore zonal winds at 50 and 25 hPa (Naujokat, 1986). The MEI is determined using a combination of the principal component analysis of sea level pressure, sea

surface temperature, zonal and meridional surface winds, surface air temperature and cloudiness as described by Wolter et al. (1998, 2011). Figure 14(a) shows the time series of the QSI and the MEI, along with the vertical velocities averaged over 60-30 hPa derived using raw model fields. As can be seen, these vertical velocities are clearly correlated with the QSI but also show a strong relationship with the MEI in some years. Figure 14(b-e) displays the results of fitting the model described by equation 10 to the raw (b) and satellite-sampled (c-e) derived vertical velocities. The time offsets were fitted using the raw

model fields and then imposed onto the satellite-sampled data. We do not fit a modeled seasonal cycle, such as the one described by equation 5, because the methodology used suppresses the seasonal cycle (Flury et al., 2013). As shown, this model is able to capture most of the variability in the derived vertical velocities. The fits are primarily driven by the QSI, with MLS sampling overestimating its influence by 3.8% (the differences in $q$ in the equations shown in Figure 14), while HALOE and ACE-FTS sampling underestimate it by 30.7 and 31.5%, respectively. The impact of the sampling is more pronounced for the MEI (the

differences in $e$), with MLS, HALOE and ACE-FTS underestimating its influence by 11, 64 and 122%. We emphasize that these sampling-induced offsets to the strength of the modulation effects of the QBO and ENSO on the circulation are only applicable to CMAM30-SD fields. These fields may not accurately represent the stratospheric tropical vertical velocities and, consequently, the actual sampling offsets could be different. As such, they should be considered only as potential biases.



The changes in tropical upwelling associated with QBO and ENSO assessed here have been shown to alter $O_3$ transport to the midlatitude lower stratosphere and to account for approximately half the interannual variability in midlatitude tropospheric $O_3$ (Neu et al., 2014). It has been hypothesized that this observed relationship between stratospheric upwelling changes and changes in tropospheric $O_3$ may provide an emergent constraint on the tropospheric $O_3$ response to long-term strengthening

of the circulation associated with greenhouse gas increases. If so, accurate quantification of the variability in tropical vertical velocities is crucial to reducing uncertainties in estimating this response.

## 6  Summary

In this paper we evaluate the effect of orbital sampling on satellite measurements of stratospheric temperature and several trace gases. In particular, we quantify the impact of sampling in terms of the sampling bias. To illustrate the impact of orbital

sampling on the outcome of representative atmospheric studies, we also quantify the induced differences in the inferred magnitude of trends and their detectability, as well as the induced differences in derived tropical vertical velocities. We calculate these sampling-induced artifacts by interpolating CMAM30-SD model fields (used as a proxy for the real atmosphere) to the real sampling patterns of three satellite instruments —Aura MLS, HALOE and ACE-FTS— to allow us to compare a dense uniform sampling pattern characteristic of limb emission sounders to the coarse non-uniform sampling patterns characteristic

of solar occultation instruments.

The results suggest that overall:

- Coarse non-uniform sampling patterns, such as the ones from HALOE and ACE-FTS, can introduce sampling biases about an order of magnitude greater than those from dense uniform sampling patterns, such as the one from MLS. For example, we found a temperature maximum sampling bias of about 10 K compared to 1 K, and $H_2O$ maximum sampling

biases as large as 5% as opposed to less than 1%, in the middle stratosphere. These results corroborate the results of Toohey et al. (2013) and Sofieva et al. (2014).

- Dense uniform sampling patterns accurately reproduce the magnitude of the model trends with only small errors. Records based on such sampling patterns will require the same number of years as when using the raw model fields, that is to say, trend detection is limited only by the natural variability. In contrast, coarse non-uniform sampling patterns may

introduce non-negligible errors in the inferred magnitude of trends, with considerably more years of data thus required to conclusively detect a given trend. This is because the sparse sampling leads to an increase in the standard deviation of the total noise in the time series. For example, for near-global temperature trends (60°S-60°N) at 10 hPa, HALOE and ACE-FTS sampling patterns artificially bias the trend estimates by about −10 and 25%, respectively. Also, at 1 hPa, the pressure level where the strongest temperature trend was found in CMAM30-SD, an MLS sampling pattern will

require 15 years to detect this particular trend, while the HALOE and ACE-FTS sampling will require 25 and 30 years, respectively.





- Coarse non-uniform sampling patterns may lead to an over or underestimation of the modulation effects of the controlling mechanisms of the tropical vertical velocities. For example, with respect to CMAM30-SD estimates, HALOE and ACE-FTS sampling patterns underestimate the QBO modulation strength by 30.7 and 31.5%, and the ENSO modulation strength by 64 and 122%, respectively. Dense uniform sampling patterns are considerably better suited to deriving tropical vertical velocities; for example, MLS sampling only overestimates the QBO influence by 3.8% and underestimates the ENSO influence by 11%.

Stratospheric changes such as an increase in the circulation and trends in temperature and $O_3$ are signatures of greenhouse gas warming and stratospheric $O_3$ recovery. Thus, our ability to accurately measure these changes is crucial for detecting anthropogenic influences on climate.

*Acknowledgements.* The research described in this paper was carried out by the Jet Propulsion Laboratory, California Institute of Technology, under contract with the National Aeronautics and Space Administration. We thank David Plummer of Environment Canada for his assistance in obtaining the CMAM30-SD dataset.



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





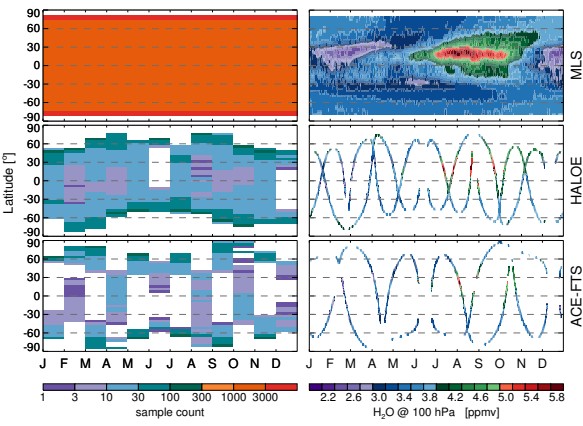

**Figure 1.** (left) Monthly sampling counts for MLS, HALOE and ACE-FTS, in 4° latitude bins. Note the non-uniform colorbar increments. (right) Zonal mean water vapor at 100 hPa as sampled by MLS, HALOE and ACE-FTS for individual days. White regions denote a lack of measurements.

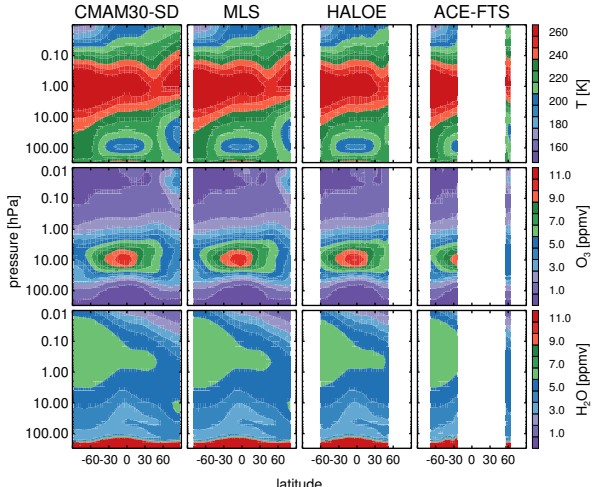

**Figure 2.** January 2005 zonal means as a function of pressure for Temperature, $O_3$ and $H_2O$ (top to bottom) in 4° latitude bins. Left column is raw CMAM30-SD model fields, other columns are CMAM30-SD as sampled by MLS, HALOE and ACE-FTS (left to right), respectively. White regions denote a lack of measurements.





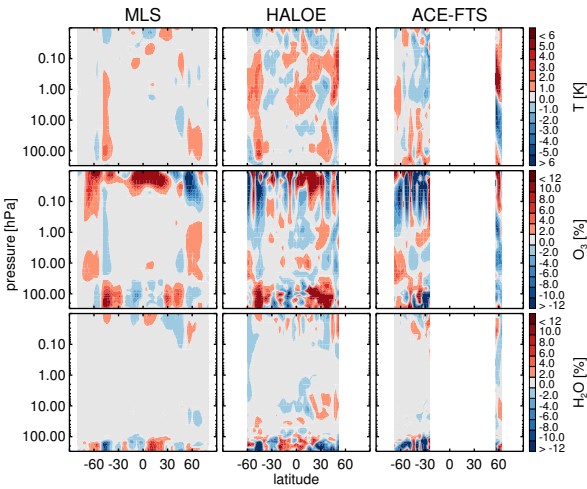

**Figure 3.** January 2005 sampling bias as a function of latitude and pressure for Temperature, $O_3$, and $H_2O$ (top to bottom) as measured using MLS, HALOE and ACE-FTS sampling patterns (left to right). White regions denote a lack of measurements.

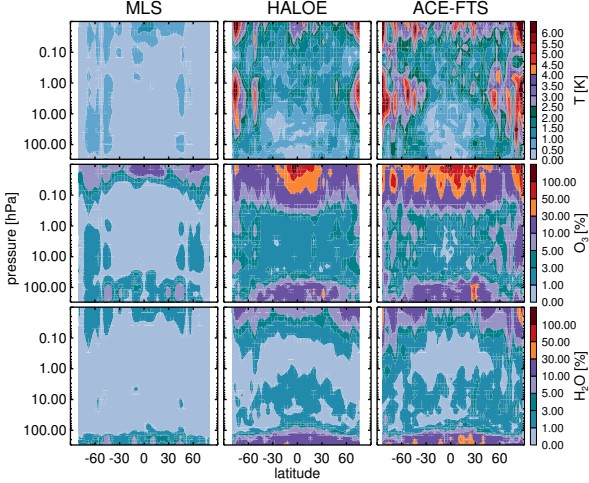

**Figure 4.** Root mean square sampling bias for 2005 as a function of latitude and pressure for Temperature, $O_3$, and $H_2O$ (top to bottom) as measured using MLS, HALOE and ACE-FTS sampling patterns (left to right). White regions denote a lack of measurements.





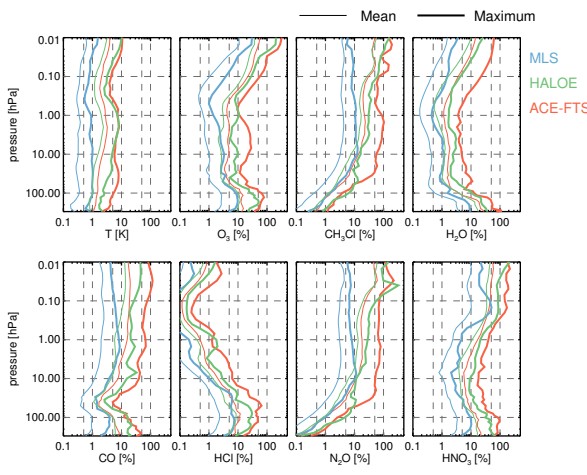

**Figure 5.** Mean (thin lines) and maximum (thicker lines) RMS sampling bias over all latitudes for 2005 as a function of pressure for Temperature, $O_3$, $CH_3Cl$, $H_2O$, CO, HCl, $N_2O$ and $HNO_3$. The vertical grid indicates values of 0.5, 1, 5, 10, 50, 100 in either K or %.

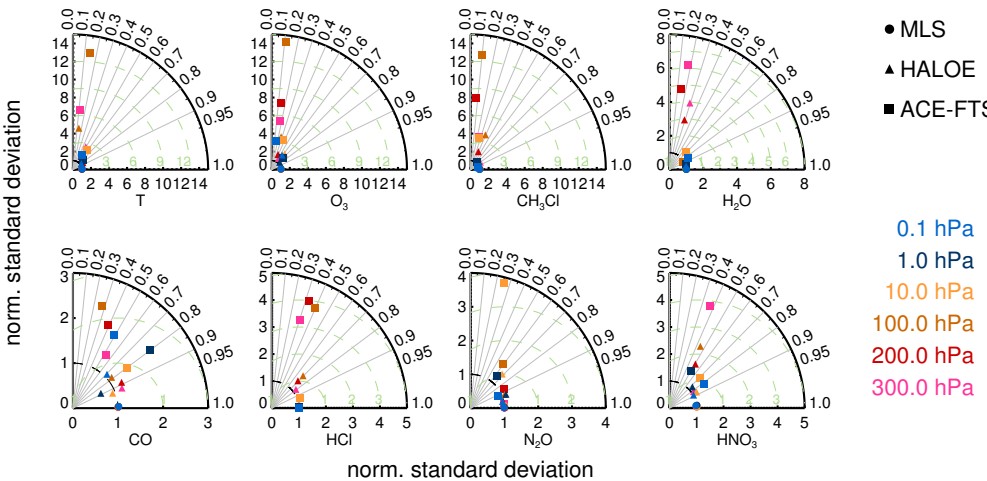

**Figure 6.** Taylor diagrams showing near-global (60°S to 60°N) long-term (1979–2012) trend comparisons between the raw (the reference point at (1,0)) and the satellite-sampled data at different pressure levels. The green contours indicate the normalized RMS difference values.





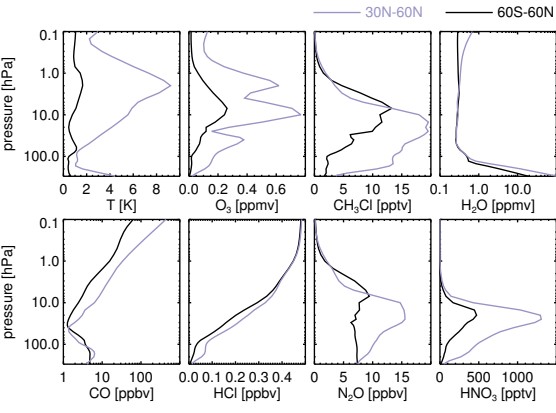

**Figure 7.** Raw model standard deviations used to normalize the Taylor diagrams shown in Figure 6 and in Figure 10. Note that $H_2O$ and CO are shown using a logarithmic scale.

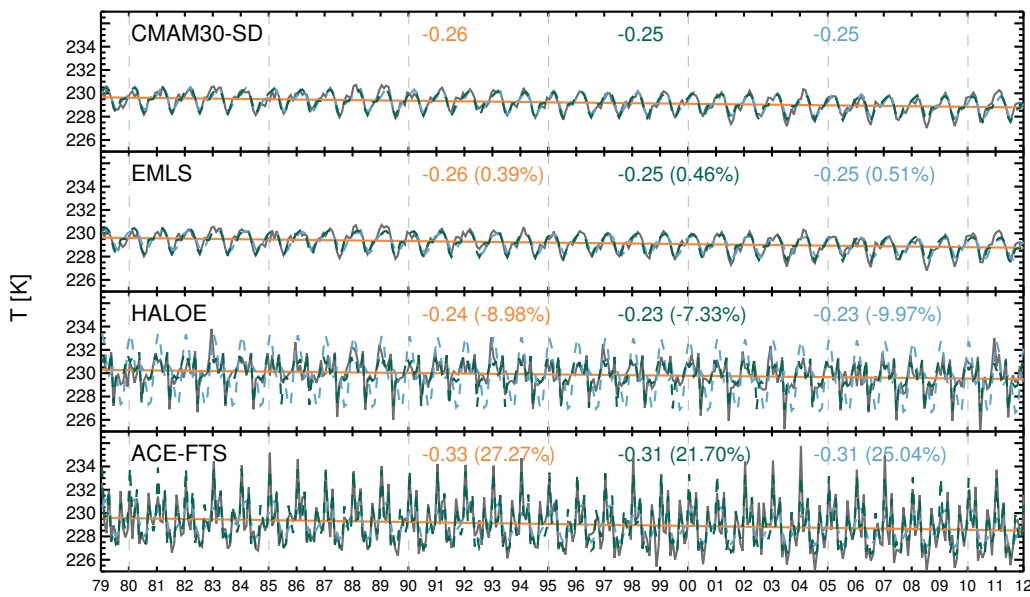

**Figure 8.** Time series of near-global (60°S to 60°N) temperature at 10 hPa for the raw and satellite-sampled data (gray lines). Orange lines display the trend computed using the linear fit, green lines show the climatological seasonal cycle imposed upon a long-term trend and light blue lines show the model computed using equation 4. The trend (K decade$^{-1}$) computed using each method is specified in each subplot; in brackets we show the percentage difference in trend magnitude with respect to the trend found using the raw model data.





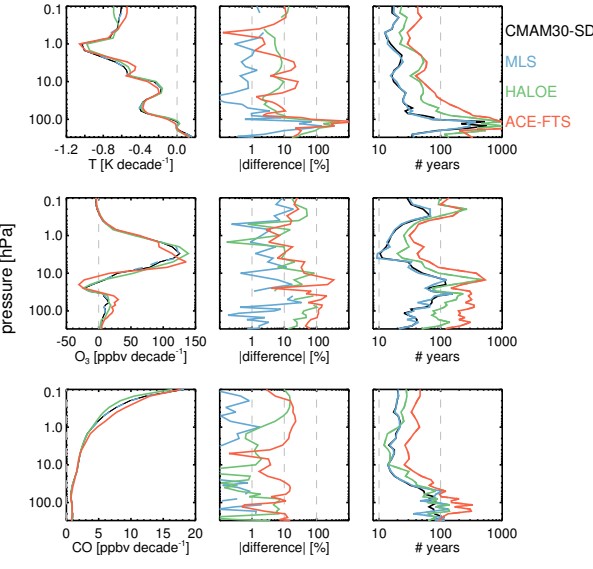

**Figure 9.** (left) Near-global (60°S to 60°N) long-term (1979–2012) trends computed using equation 4 for Temperature, $O_3$ and CO. (middle) Percentage difference in the inferred magnitude of the trends when computed using various satellite-sampled data with respect to the one computed using the raw model fields. (right) Number of years required to detect such trends.

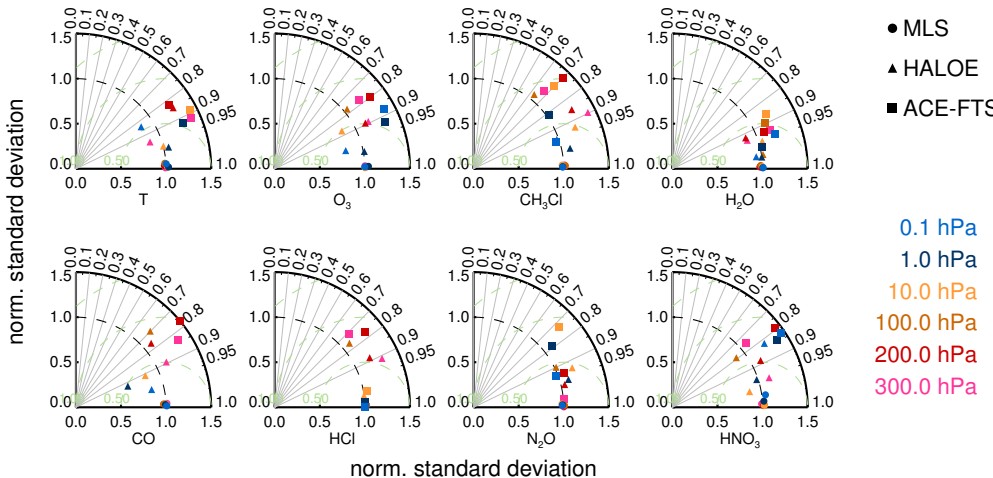

**Figure 10.** As figure 6 but for 30°N to 60°N.





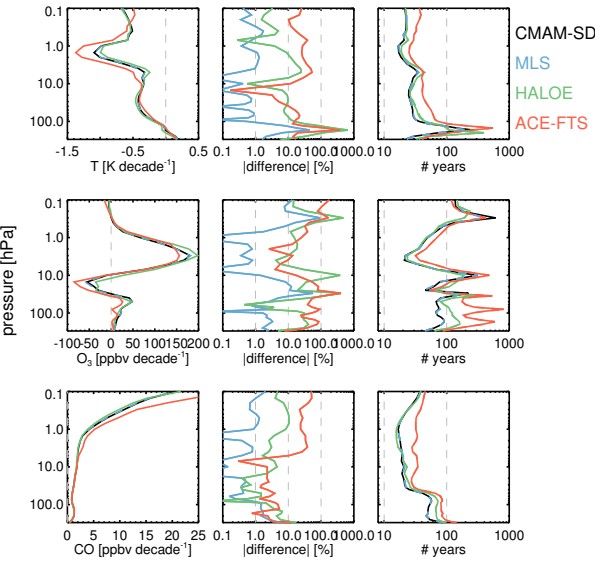

**Figure 11.** As figure 9 but for $30°$N to $60°$N.

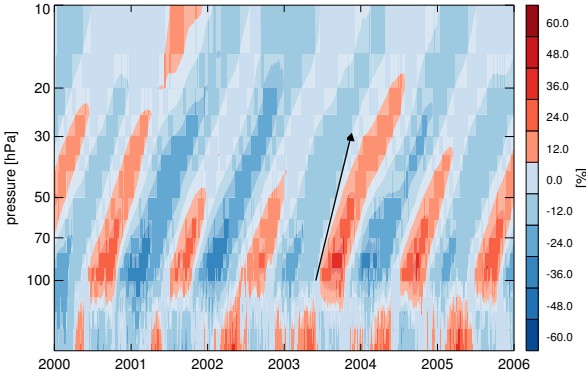

**Figure 12.** The atmospheric tape recorder (zonal mean water vapor anomalies in the tropics, in this case for CMAM30-SD raw model fields) displays a clear signal of the large-scale upward transport as indicated by the arrow. The slope of this arrow, which is derived from the propagation speed of the water vapor anomalies, represents the average tropical upwelling velocity for $8°$S-$8°$N. This subset of years is shown as an example; other years are similar.





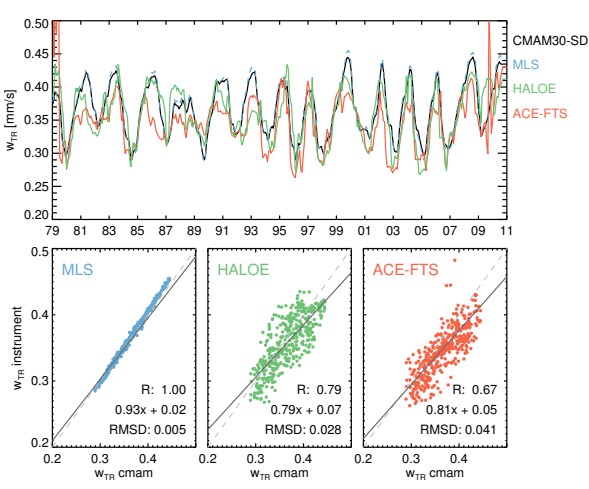

**Figure 13.** (top) $w_{TR}$ (monthly vertical velocities) derived using daily time-correlations of the water vapor tape recorder at different pressure levels from the raw CMAM30-SD data as well as the satellite-sampled data. $w_{TR}$ derived using the raw model fields and MLS-sampled data are almost identical. The pressure levels averaged are 30, 40, 50 and 60 hPa. (bottom) $w_{TR}$ scatterplots [mm/s] for MLS, HALOE and ACE-FTS sampling, respectively, versus the velocities derived using raw model fields.





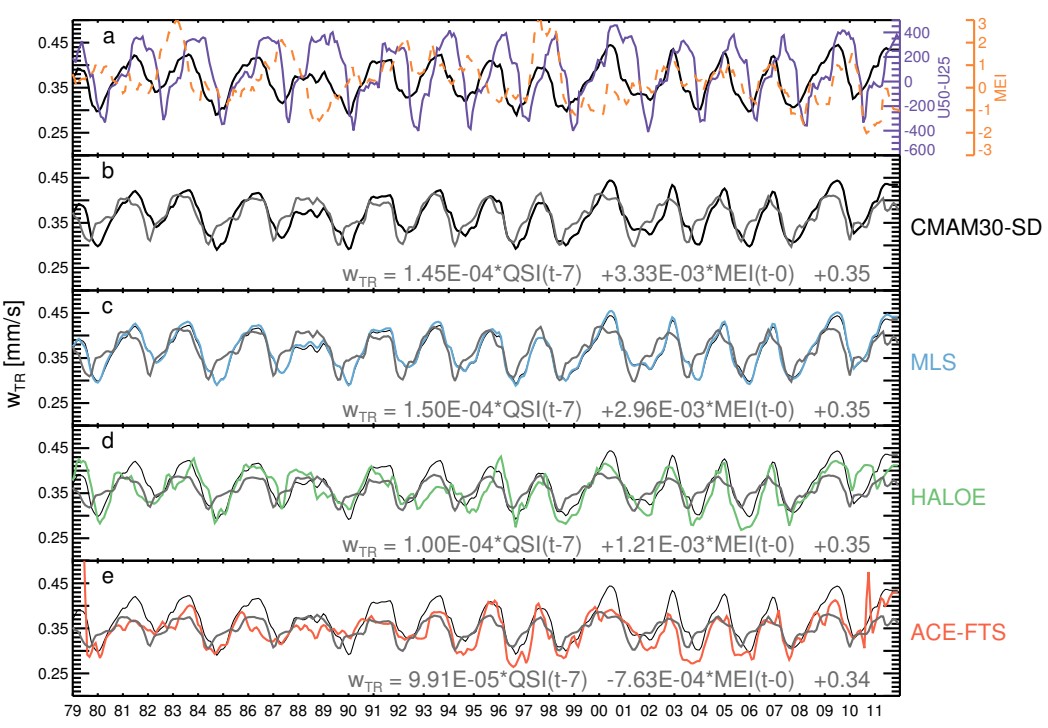

**Figure 14.** (a) Time series of $w_{TR}$ (mean monthly vertical velocities averaged over 30, 40, 50 and 60 hPa) derived using CMAM30-SD raw data (black), the quasibiennial oscillation (QBO) shear index (QSI - purple) and the Multivariate ENSO index (MEI - orange dashed line). (b) Time series of $w_{TR}$ for the raw model fields (black) as well as the model fit described by equation 10 (gray). (c-e) Time series of $w_{TR}$ for satellite-sampled data (color coded). The thin black line displays the same $w_{TR}$ derived using raw model fields (black line in panel b) for ease of comparison with the satellite-sampled ones. The model fit for each of the satellite-sampled $w_{TR}$ values, described by equation 10, is shown in gray for each of these time series.