# Peer review of "Case Studies of the Impact of Orbital Sampling on Stratospheric Trend Detection and Derivation of Tropical Vertical Velocities: Solar Occultation versus Limb Emission Sounding"

_Atmospheric Chemistry and Physics, 2016_

## Referee Comment (RC1) · Anonymous Referee #2 · 17 May 2016

This paper analyzes the impact of orbital sampling on remotely sensed data and investigates implications of related sampling biases and drifts. Although this is a methodical and partly technical paper, I recommend publication in ACP rather than any of the more technical journals, because the readership addressed are scientists working in atmospheric sciences rather than scientists working in more technical fields.

The methodology chosen is adequate; results are interesting; the presentation is concise without any unnecessary length but still complete; the authors have taken care to put their work adequately into the context of existing work. I recommend publication in

[Figure]

ACP and I have only a list of minor comments which may help to improve the paper.

p.2 l24-32: While I usually insist that results must be summarized in the abstract (as correctly done in this paper), I am not sure if it is adequate to summarize the results already in the introduction. Usually it is announced in the introduction what will be investigated, without anticipating the results.

p3 l18: It should probably read "long-term $O_3$ trends" here?

p4 l4: I miss some information about the HALOE antitude resolution here.

p4 l9: Same for ACE-FTS.

p4 l13: Same for MLS.

p4 l21: It might not even be necessary to assume that the vertical resolutions are the same. If the true fields are smooth enough in the vertical domain, even different vertical resolutions cause only marginal differences. Possibly the assumption of equal vertical resolutions can be challenged, and since a much weaker assumption (that all vertical resolutions are good enough to resolve the fields) might be sufficient, I suggest to formulate the argument with this weaker assumption.

p5 l12-22: Some discussion of the distributions of the biases might be useful. Where are the largest biases and why? Is this consistent with Toohey et al.?

p4 l22: On page 3 it is not stated how fine the vertical sampling of the model is in the lower and middle stratosphere but I guess it is much better than the vertical resolution of ACE-FTS, which is often reported to be around 3 km, if I remember correctly. Unfortunately to my knowledge neither for ACE-FTS nor for HALOE averaging kernels are available. But if you smooth the model fields using a triangular or Gaussian function of 3 km full width at half maximum, and if the model fields change only marginally by this, then you are on the safe side. At least Fig 2. suggests that there seem to exist no major vertical resolution related problems.

p5 l25 Temperature: why capital "T"?

p6 l4: RMS and standard deviation are often not adequately distinguished, and I am not sure if **centered** RMDd is a widely known concept. I would not mind to see these terms defined by equations here.

p6 l5 "everything" sounds a bit vague. "all data" or something similar would sound a bit more "intellectual" but would say essentially the same.

p6 l15 shows the raw model (remove one "the") .

p7 l1: It would be interesting to know if the autocorrelation is assumed constant over the entire time series or if it has som time-dependence in itself.

p7 l4: It would be interesting to see if the sampling artefacts somehow show up as autocorrelation. Can we learn anything about this by analysis of the $\phi$ values of the various instruments?

p7 l10: Trends in state variables can have latitudinal structure (see, e.g. Eckert et al., ACP, 2014, for ozone). I do not challenge the adequacy of trend analysis in more global terms but a caveat about latitudinal structure might be useful.

p7 l21-23: I am not happy with the term "noise" here, because this can easily be understood as "measurement noise". But here the residuum is meant, which contains real physical effects, like QBO or whatever. I suggest "The autocorrelation of the residual between the data points and the trend model" ... "The standard deviation of the residual, which corresponds ..."

l7 l25: I do not like the term "model" here, because there are so many models (trend models, CMAM, etc). I would prefer a more specific term here.

p8 l10-12: The dichotomy "solar occultation vs. microwave emission" does not exist. The HALOE and ACE-FTS data gaps due to clouds are due to the spectral region (IR), not to the measurement geometry (occultation). The IR instrument MIPAS, e.g., has

also data gaps due to clouds although it is an emission instrument. The statement made here is certainly correct (except for the dichotomy-like wording) but it seems to me to have nothing to do with orbital sampling. I think that the discussion of this issue detracts from the main issue of this paper. If you want to discuss sampling artefacts due to clouds, the fact that IR sounders have a sampling bias towards the cloud-free atmosphere would be interesting and deserves an investigation, particularly in the context of water vapor, but this is clearly beyond the scope of this paper.

Sect 4 general: Unless I have got things completely wrong, the following question remains unanswered: MLS provides both **denser** sampling and **more uniform** sampling. Both characteristics contribute to the better MLS trends. Which of the sampling attributes (data density or uniform sampling) is the primary cause of the better MLS trends? Could analysis of the $\phi$ values help to find this out?

If I haven't missed anything, the data points have equal weight in the trend fit. Is this adequate? Since orbits converge with higher latitudes, each data point represents a smaller area at higher latitudes. Thus, high latitudes are over-represented if no area-weighting is applied. This, however, is a kind of sampling bias in itself.

p9 l13 ff: there appear a lot of abbreviations and acronyms (QBO, ENSO, QSI, MEI). Please make sure that they all are defined. Some of them are but I have not checked all of them.

p10 l26: Here the better performance of instruments with MLS sampling pattern is attributed to the data density. This is intuitive but I am not sure if it has really been shown that this is not due to uniformity. The $\phi$ is not necessarily the same for the instruments under investigation.

p11 l7: This reads a bit as if the increase in the circulation is a fact but to my knowledge it is an expectation based on model calculations. To my knowledge the increase of circulation has not yet been empirically confirmed, at least not beyond any doubt. I would prefer "such as a possible increase in the circulation". For my personal taste,

the last three lines are a bit too general and I would prefer a concluding remark which is more focused on the sampling issue.

p11 l10: Finally a fussy one: (Do not take it too seriously because as a non-native English speaker I may be wrong here) It sounds funny to me that a laboratory does the work. I thought it is the researchers who do the work. The researchers are listed in the authors' list, along with their affiliations. Thus it seems redundant to me to mention this again in the acknowledgements.

Fig Cap 5: "either K or %" sounds a bit vague. Better: ...as a function of pressure for temperature (in Kelvin) and $O_3$.... (in %). Again: why capital "T" for temperature"?

---

## Referee Comment (RC2) · Anonymous Referee #1 · 24 May 2016

General Comments:

This study investigates the impact of satellite instrument sampling patterns on the derived monthly mean, latitude resolved data set produced from such data, and extends previous studies by assessing the impact of sampling on derived quantities including trends and vertical velocities. This work is a valuable addition to the field, and should add to the understanding and appreciation of sampling on atmospheric studies. I find the subject matter therefore appropriate for ACP. The manuscript is generally well written, although some important details regarding the methods should be expanded upon

(see below). I have a few comments which should be addressed before publication.

1. Are the trend and vertical velocity estimates based on solar occultation (SO) sampled fields really inaccurate, or rather are they simply more uncertain than those from the denser sampled fields? It is clear that the SO sampling adds noise to the time-series. If this noise is relatively random, you might assume that the derived quantities like trends would be simply more uncertain. To say that the sampling leads to inaccurate derived quantities would require comparing not just the estimated trends and vertical velocities, but the error of those estimates. For example, it's not hard to imagine that the cloud of points in Fig 13 for the HALOE and ACE-FTS sampled fields might be consistent with the 1:1 line, which would be reflected by an estimated slope whose uncertainty interval contained 1. Therefore, to say that the SO sampling leads to biased trends and vertical velocities requires calculation of the errors in those trend and velocity estimates.

2. ACE-FTS samples the tropics in only 4 months of the year. It is therefore quite a stretch to compute full timeseries of monthly mean tropical upwelling from such sparse data, and I would be quite wary of any study that attempted this with real data. Very little detail is given on how the authors here fill in the missing months, and I am suspicious that the large error that they conclude results from the sparse sampling could be dominated by the interpolation used. A fairer method would be to compute the tropical upwelling only for months in which the SO instruments actually sample the full 8°S-8°N band. This would rule out the chance that the result here is produced by the interpolation. This might make the more complex analysis of Sec. 5 more difficult, or even impossible when applied to the ACE-FTS data. But, it could be argued that if the analysis doesn't work when applied to such sparse data, then it doesn't necessarily need to be proven that it doesn't work on highly interpolated data.

Specific Comments:

p2, l24: accuracy, or precision? See general comment 1.

[Figure]

p2, l34: the impact of sampling on the estimation of long-term trends

p3, l18: should this be actually O3 trends?

p3, l20: spatial->horizontal

p4, l21: focus "on"

p5, eq 1: could be written and explained clearer. What is y? Is x the data (as written), or a placeholder variable for "r" and "s"?

p5, l12: Should make clear here that 2005 refers to the model year, not the sampling pattern (which is assumed constant over all years).

p5, l18 (and elsewhere): why capital T in temperature?

p6, l3-4: the quantities shown in a Taylor diagram seem to compare the variability of two data sets, not really the trends in the data sets. I guess that two data sets could have the same trend, but perform poorly on these diagnostics. (I bring this up because the paragraph, and the section in general very much focuses on long-term trends, rather than the short-term variability that the correlation coefficient measures the agreement of.)

p6, l12: how are 60°S-60°N means calculated from the sparse SO fields if the full latitude range is not actually sampled in a month? The simplest method is taking the 60°S-60°N model mean, and subtracting it from the mean of whatever is available from the satellite sampled field. This creates large differences for sparsely sampled data. Another method, which is perhaps more fair to the sparser data, is to first take the difference between the model and sampled fields for each latitude, and then average the difference, which gives then the average difference over the latitudes where the satellite samples. Whatever method is used here should be clearly described, and the implications of the choice discussed.

p7, l1: How is $\sigma_\epsilon^2$ estimated? It seems clear that the sampling affects the random error

in the time series, is this accounted for in the fitting procedure?

p7, l20: What values are used for $\phi$, and how is it estimated?

p9, l27: it's not just the sampling that biases the trend, also important is the procedure of calculating 60S-60N means when the coverage is incomplete. A subtle point, but I think worth repeating.

Fig 13: The MLS-sampled scatter plot of Fig 13 looks surprising: to the naked eye, the points seem to lie along the 1:1 line for the smaller $w_{TR}$ values, and look to lie above the 1:1 line for larger $w_{TR}$ values (this also appears visible in the timeseries plot, where the MLS sampled field appears to give larger values at the yearly peak on $w_{TR}$). However, the fit gives, surprisingly, a slope less than 1. In any case, a short description of the method used to produce the linear fits should be included, as there are a number of methods possible, which give different answers and depend on different sets of assumption (see e.g., Isobe et al., 1990).

References:

Isobe, T., Feigelson, E. D., Akritas, M. G. and Babu, G. J.: Linear regression in astronomy., Astrophys. J., 364, 104, doi:10.1086/169390, 1990.

---

## Author Comment (AC2) · 13 Jun 2016

We thank the reviewer for his/her comments. Below are our responses in blue.

**1   Reviewer 1 comments**

General Comments:

[Figure]

This study investigates the impact of satellite instrument sampling patterns on the derived monthly mean, latitude resolved data set produced from such data, and extends previous studies by assessing the impact of sampling on derived quantities including trends and vertical velocities. This work is a valuable addition to the field, and should add to the understanding and appreciation of sampling on atmospheric studies. I find the subject matter therefore appropriate for ACP. The manuscript is generally well written, although some important details regarding the methods should be expanded upon(see below). I have a few comments which should be addressed before publication.

1. Are the trend and vertical velocity estimates based on solar occultation (SO) sampled fields really inaccurate, or rather are they simply more uncertain than those from the denser sampled fields? It is clear that the SO sampling adds noise to the timeseries. If this noise is relatively random, you might assume that the derived quantities like trends would be simply more uncertain. To say that the sampling leads to inaccurate derived quantities would require comparing not just the estimated trends and vertical velocities, but the error of those estimates. For example, it's not hard to imagine that the cloud of points in Fig 13 for the HALOE and ACE-FTS sampled fields might be consistent with the 1:1 line, which would be reflected by an estimated slope whose uncertainty interval contained 1. Therefore, to say that the SO sampling leads to biased trends and vertical velocities requires calculation of the errors in those trend and velocity estimates.

That is correct for noise relatively random, however the noise is determined by the sampling pattern of the geophysical parameter, that is why the trends and tropical velocities are different, which is what we are investigating in this study. If the noise due to the sampling was random, the trends and tropical velocities will be the same as in CMAM30-SD but noisier, as the reviewer point out, but they are not. Further, if something has big error bars that encompass the "1" does not imply that it cannot be biased, in fact, it can be bias and more uncertain, as it is pressumably the case for the

SO.

2. ACE-FTS samples the tropics in only 4 months of the year. It is therefore quite a stretch to compute full timeseries of monthly mean tropical upwelling from such sparse data, and I would be quite wary of any study that attempted this with real data. Very little detail is given on how the authors here fill in the missing months, and I am suspicious that the large error that they conclude results from the sparse sampling could be dominated by the interpolation used. A fairer method would be to compute the tropical upwelling only for months in which the SO instruments actually sample the full 8S-8N band. This would rule out the chance that the result here is produced by the interpolation. This might make the more complex analysis of Sec. 5 more difficult, or even impossible when applied to the ACE-FTS data. But, it could be argued that if the analysis doesn't work when applied to such sparse data, then it doesn't necessarily need to be proven that it doesn't work on highly interpolated data.

To compute the tropical upwelling the method correlate adjacent levels using a time lag of a year. Hence, an entire timeseries is needed. See page 8 line 27 to 31 of the original manuscript. The reviewer is correct to point out that for ACE-FTS data we are mainly doing the analysis in highly interpolated data. The following sentence will be added to emphasis this: For ACE-FTS gaps are filled in January, March, May, July, September, November and December, when no measurements are made over the tropics (8°S to 8°N), **as such, we are applying the analysis to highly interpolated data.**

Specific Comments:

p2, l24: accuracy, or precision? See general comment 1. We meant accuracy but this sentence will be deleted from the final manuscript due to a comment from referee 2.

p2, l34: the impact of sampling on the estimation of long-term trends ok

p3, l18: should this be actually O3 trends? It will be changed to: long term $O_3$ loss

p3, l20: spatial- horizontal ok

p4, l21: focus "on" ok

p5, eq 1: could be written and explained clearer. What is y? Is x the data (as written), or a placeholder variable for "r" and "s"? The text will be changed to: where $N$ is the total number of points, $y$, belonging to a latitude bin $l$, and x is a placeholder variable for either the raw data, denoted by the superscript $r$, or the sampled data, denoted by the superscript $s$.

p5, l12: Should make clear here that 2005 refers to the model year, not the sampling pattern (which is assumed constant over all years). the sentence will be changed to: Figure 3 shows examples of the sampling biases for temperature, O3 and H2O for January 2005 CMAM30-SD data.

p5, l18 (and elsewhere): why capital T in temperature? it will be change for lowercase

p6, l3-4: the quantities shown in a Taylor diagram seem to compare the variability of two data sets, not really the trends in the data sets. I guess that two data sets could have the same trend, but perform poorly on these diagnostics. (I bring this up because the paragraph, and the section in general very much focuses on long-term trends, rather than the short-term variability that the correlation coefficient measures

the agreement of.) That is correct, two datasets can have the same trend but perform poorly in a Taylor diagram due to either a lack of correlation or different standard deviations. Both cases will result in an increase in the number of years required to detect such a trend.

p6, l12: how are 60S-60N means calculated from the sparse SO fields if the full latitude range is not actually sampled in a month? The simplest method is taking the 60S-60N model mean, and subtracting it from the mean of whatever is available from the satellite sampled field. This creates large differences for sparsely sampled data. Another method, which is perhaps more fair to the sparser data, is to first take the difference between the model and sampled fields for each latitude, and then average the difference, which gives then the average difference over the latitudes where the satellite samples. Whatever method is used here should be clearly described, and the implications of the choice discussed.

The means were computed simply be averaging whatever was available. The difference were computed using the first method described, the second one will not be representative of the 60S-60N difference. We will add the following sentence: Means were computed averaging all the data available between 60N and 60S with no effort to use only latitudes where the satellites sampled. This was performed to show the representativeness of near global trends.

p7, l1: How is $\sigma_\varepsilon^2$ estimated? $\sigma_{varepsilon}^2$ is computed from $\sigma_N$ using equation 8. It seems clear that the sampling affects the random error in the time series, is this accounted for in the fitting procedure? Not directly in the fitting procedure, $\sigma_N$ is simply the standard deviation of the residuals between the data points and the trend model, hence, its magnitude is affected by the sampling artifacts. The sentence in page 7 line 21 of the original document will be changed to: (1) $\phi$ the autocorrelation of the residual between the data points and the trend model computed following Tiao et al. (1990)(2)

$\sigma_N$ the standard deviation of the residual, which corresponds...

p7, l20: What values are used for $\phi$, and how is it estimated? $\phi$ is estimated following the Tiao et al (1990) appendix A. We will add to the text (see comment above): computed following Tiao et al. (1990). We will also add: $\phi$ is computed for the raw as well as the satellite-sampled data.

p9, l27: it's not just the sampling that biases the trend, also important is the procedure of calculating 60S-60N means when the coverage is incomplete. A subtle point, but I think worth repeating. See response above.

Fig 13: The MLS-sampled scatter plot of Fig 13 looks surprising: to the naked eye, the points seem to lie along the 1:1 line for the smaller wTR values, and look to lie above the 1:1 line for larger wTR values (this also appears visible in the timeseries plot, where the MLS sampled field appears to give larger values at the yearly peak on wTR). However, the fit gives, surprisingly, a slope less than 1. In any case, a short description of the method used to produce the linear fits should be included, as there are a number of methods possible, which give different answers and depend on different sets of assumption (see e.g., Isobe et al., 1990). the reviewer is correct on his/her depiction of the figure. However, it is the density of points on the low side (for smaller wTR values) what is driving the slope. We will add the following sentence in the caption: Note that the MLS slope is driven by the vast number of points associated with small wTR values. We will also state that the linear fit is an ordinary least-squares regression.

References:
Isobe, T., Feigelson, E. D., Akritas, M. G. and Babu, G. J.: Linear regression in astronomy., Astrophys. J., 364, 104, doi:10.1086/169390, 1990.

---

## Author Response (AR1)

**Author's response**

August 11, 2016

We thank the reviewers for their comments. Below are our responses in blue.

**1 Reviewer 1 comments**

General Comments:

This study investigates the impact of satellite instrument sampling patterns on the derived monthly mean, latitude resolved data set produced from such data, and extends previous studies by assessing the impact of sampling on derived quantities including trends and vertical velocities. This work is a valuable addition to the field, and should add to the understanding and appreciation of sampling on atmospheric studies. I find the subject matter therefore appropriate for ACP. The manuscript is generally well written, although some important details regarding the methods should be expanded upon(see below). I have a few comments which should be addressed before publication.

1. Are the trend and vertical velocity estimates based on solar occultation (SO) sampled fields really inaccurate, or rather are they simply more uncertain than those from the denser sampled fields? It is clear that the SO sampling adds noise to the timeseries. If this noise is relatively random, you might assume that the derived quantities like trends would be simply more uncertain. To say that the sampling leads to inaccurate derived quantities would require comparing not just the estimated trends and vertical velocities, but the error of those estimates. For example, it's not hard to imagine that the cloud of points in Fig 13 for the HALOE and ACE-FTS sampled fields might be consistent with the 1:1 line, which would be reflected by an estimated slope whose uncertainty interval contained 1. Therefore, to say that the SO sampling leads to biased trends and vertical velocities requires calculation of the errors in those trend and velocity estimates.

That is correct for relatively random noise, however the noise is determined by the sampling pattern of the geophysical parameter, that is why the trends and tropical velocities are different, which is what we are investigating in this study. If the noise due to the sampling was random, the trends and tropical velocities will be the same as in CMAM30-SD but noisier, as the reviewer points out, but they are not. Further, if something has big error bars that encompass "1", that does not imply that it cannot be biased, in fact, it can be biased and more uncertain, as is presumably the case for the SO. Also, we computed the confidence intervals of the slopes and they do not encompass 1. We will add the following line to the caption of figure 13: The slopes' confidence intervals are $\pm 0.007$, $0.06$ and $0.09$ for MLS, HALOE and ACE-FTS, respectively.

2. ACE-FTS samples the tropics in only 4 months of the year. It is therefore quite a stretch to compute full timeseries of monthly mean tropical upwelling from such sparse data, and I would be quite wary of any study that attempted this with real data. Very little detail is given on how the authors here fill in the missing months, and I am suspicious that the large error that they conclude results from the sparse sampling could be dominated by the interpolation used. A fairer method would be to compute the tropical upwelling only for months in which the SO instruments actually sample the full 8S-8N band. This would rule out the chance that the result here is produced by the interpolation. This might make the more complex analysis of Sec. 5 more difficult, or even impossible when applied to the ACE-FTS data. But, it could be argued that if the analysis doesn't work when applied to such sparse data, then it doesn't necessarily need to be proven that it doesn't work on highly interpolated data.

To compute the tropical upwelling the method correlates adjacent levels using a time lag of a year. Hence, an entire timeseries is needed. See page 8 line 27 to 31 of the original manuscript. The reviewer is correct to point out that for ACE-FTS data we are mainly doing the analysis on highly interpolated data. The following sentence will be added to emphasize this (the bold part shows the added text): For ACE-FTS gaps are filled in January, March, May, July, September, November and December, when no measurements are made over the tropics (8°S to 8°N); **thus, we are applying the analysis to highly interpolated data. Considering the degree of interpolation required, we do not recommend the use of ACE-FTS to derive tropical upwelling velocities but we include this case merely as an illustrative example.**

Specific Comments:

p2, l24: accuracy, or precision? See general comment 1. We meant accuracy but this sentence will be deleted from the final manuscript in response to a comment from referee 2.

p2, l34: the impact of sampling on the estimation of long-term trends agreed

p3, l18: should this be actually O3 trends? Agreed

p3, l20: spatial- horizontal agreed

p4, l21: focus "on" agreed

p5, eq 1: could be written and explained clearer. What is y? Is x the data (as written), or a placeholder variable for "r" and "s"? The text will be changed to: where $N$ is the total number of points, $y$, belonging to a latitude bin $l$, and x is a placeholder variable for either the raw data, denoted by the superscript $r$, or the sampled data, denoted by the superscript $s$.

p5, l12: Should make clear here that 2005 refers to the model year, not the sampling pattern (which is assumed constant over all years). the sentence will be changed to: Figure 3 shows examples of the sampling biases for temperature, $O_3$ and $H_2O$ for January 2005 CMAM30-SD data.

p5, l18 (and elsewhere): why capital T in temperature? it will be change for lowercase

p6, l3-4: the quantities shown in a Taylor diagram seem to compare the variability of two data sets, not really the trends in the data sets. I guess that two data sets could have the same trend, but perform poorly on these diagnostics. (I bring this up because the paragraph, and the section in general very much focuses on long-term trends, rather than the short-term variability that the correlation coefficient measures the agreement of.) That is correct, two datasets can have the same trend but perform poorly in a Taylor diagram due to either a lack of correlation or different standard deviations. Both cases will result in an increase in the number of years required to detect such a trend. We will add the following sentence after the discussion of the Taylor diagrams for the extratropics (Figure 10): Two variables can have similar trends but still perform poorly in Taylor diagrams due to either a lack of correlation or different standard deviations. In both cases, this will impact $\sigma_N$, resulting in an increase in the number of years required to statistically detect such a trend.

p6, l12: how are 60S-60N means calculated from the sparse SO fields if the full latitude range is not actually sampled in a month? The simplest method is taking the 60S-60N model mean, and subtracting it from the mean of whatever is available from the satellite sampled field. This creates large differences for sparsely sampled data.
Another method, which is perhaps more fair to the sparser data, is to first take the difference between the model and sampled fields for each latitude, and then average the difference, which gives then the average difference over the latitudes where the satellite samples. Whatever method is used here should be clearly described, and the implications of the choice discussed.
The means were computed simply by averaging whatever was available. The difference were computed using

the first method described, the second one will not be representative of the 60S-60N difference. In addition, the second approach suggested by the reviewer is only applicable in cases where a model (or some dataset with more even latitude coverage) is available or being considered. In many previously published studies, authors have taken averages of solar occultation data (for example Brown et al. (2011)) and used them as measures of the variability over that range, an approach equivalent to the reviewer's first method (the one we employ here). We will add the following sentence: Means were computed by averaging all data available between 60N and 60S with no effort to use only latitudes where the satellites sampled. This approach was taken to show the representativeness of near-global trends.

p7, l1: How is $\sigma_\varepsilon^2$ estimated? $\sigma_\varepsilon^2$ is computed from $\sigma_N$ using equation 8.
It seems clear that the sampling affects the random error in the time series, is this accounted for in the fitting procedure? Not directly in the fitting procedure, $\sigma_N$ is simply the standard deviation of the residuals between the data points and the trend model, hence, its magnitude is affected by the sampling artifacts. The sentence on page 7 line 21 of the original document will be changed to: (1) $\phi$ the autocorrelation of the residual between the data points and the trend model computed, and assumed temporally invariant, following Tiao et al. (1990), (2) $\sigma_N$ the standard deviation of the residual, which corresponds...

p7, l20: What values are used for $\phi$, and how is it estimated? $\phi$ is estimated following the Tiao et al. (1990) appendix A. We will add to the text (see comment above): computed following Tiao et al. (1990). We will also add: $\phi$ is computed for the raw as well as the satellite-sampled data.

p9, l27: it's not just the sampling that biases the trend, also important is the procedure of calculating 60S-60N means when the coverage is incomplete. A subtle point, but I think worth repeating. See response above.

Fig 13: The MLS-sampled scatter plot of Fig 13 looks surprising: to the naked eye, the points seem to lie along the 1:1 line for the smaller wTR values, and look to lie above the 1:1 line for larger wTR values (this also appears visible in the timeseries plot, where the MLS sampled field appears to give larger values at the yearly peak on wTR). However, the fit gives, surprisingly, a slope less than 1. In any case, a short description of the method used to produce the linear fits should be included, as there are a number of methods possible, which give different answers and depend on different sets of assumption (see e.g., Isobe et al., 1990). the reviewer is correct in his/her depiction of the figure. At first we thought it was due to the smaller $w_{TR}$ values driving the slopes. However, when computing the confidence intervals for the slopes, we found a typo in the code. In the new version the slope is 1.07. We will also state that the linear fit is an ordinary least-squares regression.

References:
Isobe, T., Feigelson, E. D., Akritas, M. G. and Babu, G. J.: Linear regression in astronomy., Astrophys. J., 364, 104, doi:10.1086/169390, 1990.
Brown, A. T., Martyn P. Chipperfield, Chris Boone, Chris Wilson, Kaley A. Walker, Peter F. Bernath, Trends in atmospheric halogen containing gases since 2004, Journal of Quantitative Spectroscopy and Radiative Transfer, Volume 112, Issue 16, November 2011, Pages 2552-2566, doi: 10.1016/j.jqsrt.2011.07.005.

**2 Reviewer 2 comments**

**Summary**

This paper analyzes the impact of orbital sampling on remotely sensed data and investigates implications of related sampling biases and drifts. Although this is a methodical and partly technical paper, I recommend publication in ACP rather than any of the more technical journals, because the readership addressed are scientists working in atmospheric sciences rather than scientists working in more technical fields.

The methodology chosen is adequate; results are interesting; the presentation is concise without any unnecessary length but still complete; the authors have taken care to put their work adequately into the context of existing work. I recommend publication in ACP and I have only a list of minor comments which

may help to improve the paper.

**Comments**

p.2 l24-32: While I usually insist that results must be summarized in the abstract (as correctly done in this paper), I am not sure if it is adequate to summarize the results already in the introduction. Usually it is announced in the introduction what will be investigated, without anticipating the results. The lines detailing the results will be deleted.

p3 l18: It should probably read "long-term O3 trends" here? agreed

p4 l4: I miss some information about the HALOE altitude resolution here. This sentence will be added: The vertical resolution of this dataset is about 2.3 km.

p4 l9: Same for ACE-FTS. This sentence will be added: The vertical resolution of this dataset is about 3 km.

p4 l13: Same for MLS. This sentence will be added: The vertical resolution of this dataset varies among species; $O_3$, $H_2O$ and HCl have a $\sim 3$ km resolution in the stratosphere, and CO, $CH_3Cl$, $HNO_3$ and $N_2O$ have a 4–8 km resolution in the stratosphere [Livesey et al. 2015].

p4 l21: It might not even be necessary to assume that the vertical resolutions are the same. If the true fields are smooth enough in the vertical domain, even different vertical resolutions cause only marginal differences. Possibly the assumption of equal vertical resolutions can be challenged, and since a much weaker assumption (that all vertical resolutions are good enough to resolve the fields) might be sufficient, I suggest to formulate the argument with this weaker assumption. We will add (the bold part shows the added text): Given that our focus is on horizontal/temporal sampling, all satellite measurements are assumed to have vertical resolution comparable to that of CMAM30-SD; **however, we want to emphasize that the vertical resolution of these instruments is in general good enough to resolve the model fields.**

p5 l12-22: Some discussion of the distributions of the biases might be useful. Where are the largest biases and why? Is this consistent with Toohey et al.? Although we do not discuss Figure 3 sampling biases we do discuss the RMS sampling biases in Figure 4 (page5 line 19 of the original document) and compare them against Toohey et al. We believe that a discussion of Figure 3 sampling biases will be redundant.

p4 l22: On page 3 it is not stated how fine the vertical sampling of the model is in the lower and middle stratosphere but I guess it is much better than the vertical resolution of ACE-FTS, which is often reported to be around 3 km, if I remember correctly. Unfortunately to my knowledge neither for ACE-FTS nor for HALOE averaging kernels are available. But if you smooth the model fields using a triangular or Gaussian function of 3 km full width at half maximum, and if the model fields change only marginally by this, then you are on the safe side. At least Fig 2. suggests that there seem to exist no major vertical resolution related problems. we will modify this sentence to (the bold part shows the added text): That is, although the impact of the averaging kernels is not addressed in this study, **for the parameters studied here a 3 km averaging kernel does not significantly affect their values in the upper troposphere/stratosphere.**

p5 l25 Temperature: why capital "T"? it will be changed to lowercase

p6 l4: RMS and standard deviation are often not adequately distinguished, and I am not sure if centered RMDd is a widely known concept. I would not mind to see these terms defined by equations here. After careful consideration we decided against including more equations that might add complexity to the study, we feel that the reader could easily go to any statistic books and to Taylor (2001) for detailed descriptions. We will add the following sentence as a brief explanation of centered RMSd: Simply, the centered RMSd is the RMS of the differences between the two anomaly time series.

p6 l5 "everything" sounds a bit vague. "all data" or something similar would sound a bit more "intellectual" but would say essentially the same. as suggested, we will use "all data"

p6 l15 shows the raw model (remove one "the"). agreed

p7 l1: It would be interesting to know if the autocorrelation is assumed constant over the entire time series or if it has some time-dependence in itself. the autocorrelation is assumed constant over the entire time series, we will add: computed following Tiao et al. (1990), so the reader can see how it was estimated. Also, we will add on page 7 line 20: $\phi$ is computed for the raw as well as the satellite-sampled data.

p7 l4: It would be interesting to see if the sampling artifacts somehow show up as autocorrelation. Can we learn anything about this by analysis of the values of the various instruments? The reviewer makes an excellent point. The figures shown use the autocorrelation derived from the raw model as well as the satellite-sampled data. However, using the autocorrelation from the raw-model instead of the satellite-sampled data did not produce significantly different results. We will add the following sentence at the end of page 7 (original document): We also performed this analysis using only the autocorrelation computed for the raw model data and found no significant differences.

p7 l10: Trends in state variables can have latitudinal structure (see, e.g. Eckert et al., ACP, 2014, for ozone). I do not challenge the adequacy of trend analysis in more global terms but a caveat about latitudinal structure might be useful. That is partly why we included Figure 11. We will add the following sentence at the end of section 4: As shown, the ability to detect trends depends upon the natural variability and the correlation of the data. These in turn vary with the specific parameter as well as the location and height being studied. Studies of natural variability and autocorrelation of the data will help identify where to monitor to find more readily detectable trends, but such a study is outside the scope of this paper.

p7 l21-23: I am not happy with the term "noise" here, because this can easily be understood as "measurement noise". But here the residuum is meant, which contains real physical effects, like QBO or whatever. I suggest "The autocorrelation of the residual between the data points and the trend model" ... "The standard deviation of the residual, which corresponds ..." agreed

l7 l25: I do not like the term "model" here, because there are so many models (trend models, CMAM, etc). I would prefer a more specific term here. It will be changed to trend model (equation 4)

p8 l10-12: The dichotomy "solar occultation vs. microwave emission" does not exist. The HALOE and ACE-FTS data gaps due to clouds are due to the spectral region (IR), not to the measurement geometry (occultation). The IR instrument MIPAS, e.g., has also data gaps due to clouds although it is an emission instrument. The statement made here is certainly correct (except for the dichotomy-like wording) but it seems to me to have nothing to do with orbital sampling. I think that the discussion of this issue detracts from the main issue of this paper. If you want to discuss sampling artifacts due to clouds, the fact that IR sounders have a sampling bias towards the cloud-free atmosphere would be interesting and deserves an investigation, particularly in the context of water vapor, but this is clearly beyond the scope of this paper. We will delete that sentence.

Sect 4 general: Unless I have got things completely wrong, the following question remains unanswered: MLS provides both denser sampling and more uniform sampling. Both characteristics contribute to the better MLS trends. Which of the sampling attributes (data density or uniform sampling) is the primary cause of the better MLS trends? That is correct, that question remains unanswered and a more detailed study, presumably with fake sampling patterns, will have to be carried out. Could analysis of the values help to find this out? We do not think so, see comments about $\phi$ above.

If I haven't missed anything, the data points have equal weight in the trend fit. Is this adequate? Since orbits converge with higher latitudes, each data point represents a smaller area at higher latitudes. Thus, high latitudes are over-represented if no area weighting is applied. This, however, is a kind of sampling

bias in itself. As the reviewer points out, this will affect results primarily at higher latitudes; we only use latitudes between 60S and 60N, where we do not expect any significant differences between equal weight and fractional area weighted trends.

p9 l13 ff: there appear a lot of abbreviations and acronyms (QBO, ENSO, QSI, MEI). Please make sure that they all are defined. Some of them are but I have not checked all of them. they are defined

p10 l26: Here the better performance of instruments with MLS sampling pattern is attributed to the data density. This is intuitive but I am not sure if it has really been shown that this is not due to uniformity. The $\phi$ is not necessarily the same for the instruments under investigation. See comments about $\phi$ above. We will modify the sentence to (the bold part shows the added text): This is because the sparse **non-uniform** sampling leads to an increase ...

p11 l7: This reads a bit as if the increase in the circulation is a fact but to my knowledge it is an expectation based on model calculations. To my knowledge the increase of circulation has not yet been empirically confirmed, at least not beyond any doubt. I would prefer "such as a possible increase in the circulation". For my personal taste, the last three lines are a bit too general and I would prefer a concluding remark which is more focused on the sampling issue. We prefer to keep the discussion (with the suggested correction) to re-emphasize the motivation for and importance of the work. We will modify the sentence to include "such as a possible increase in the circulation"

p11 l10: Finally a fussy one: (Do not take it too seriously because as a non-native English speaker I may be wrong here) It sounds funny to me that a laboratory does the work. I thought it is the researchers who do the work. The researchers are listed in the authors' list, along with their affiliations. Thus it seems redundant to me to mention this again in the acknowledgements. it will be changed to: Work at the Jet Propulsion Laboratory, California Institute of Technology, was done under contract with the National Aeronautics and Space Administration.

Fig Cap 5: "either K or %" sounds a bit vague. Better: ...as a function of pressure for temperature (in Kelvin) and O3.... (in %). Again: why capital "T" for temperature" ? agreed, the new sentence will be: Mean (thin lines) and maximum (thicker lines) RMS sampling bias over all latitudes for 2005 as a function of pressure for temperature (in Kelvin), and $O_3$, $CH_3Cl$, $H_2O$, CO, HCl, $N_2O$ and $HNO_3$ (in %). The vertical grid indicates values of 0.5, 1, 5, 10, 50, 100.

---

## Author Response (AR2)

**Response to reviewer**

August 29, 2016

We thank the reviewer for his/her comments. Below are our responses in blue.

**1 Reviewer comments**

The authors have carefully considered and addressed the comments of the initial review. I have only a few remaining quite minor comments for the authors to consider before publication.

P2, l14: Since the previous paragraph discusses prior work, it would be helpful if the first sentence introducing the present study highlighted what is new about this study. As is, a hasty reader might think that this is the first study to evaluate the impact of sampling bias on climatologies of MLS, HALOE and ACE-FTS, which is not the case.

P2L14 will be changed from "In this study we evaluate the impact of the Aura Microwave..." to "In this study we **further** evaluate the impact of the Aura Microwave..."

We believe that P2L20 clearly states what is new about this study: Our study has two purposes: (1) We expand upon previous studies by quantifying the sampling bias of these instruments affecting measurements of upper tropospheric and stratospheric temperature and trace gas species. (2) We investigate how differences in data coverage may affect the outcome of two illustrative atmospheric studies: trend detection and quantification of tropical vertical velocities.

P6, l16: Here, and in the figure caption, and elsewhere, the Taylor diagrams are introduced as a metric of the "trends" of the different data sets. Given that two datasets could have similar trends but show very poor agreement on a Taylor diagram (as now stated), it seems that Taylor diagrams are not really a way to assess or compare trends. I suggest nothing more than a change of word use here (and elsewhere) when introducing the Taylor diagrams.

"Trends" will be changed to "patterns" as used by Taylor, 2001.

Figure 13 caption: It would be helpful to know whether the confidence intervals given here 1 or 2 sigma.

These confidence intervals are 95% confidence intervals. The caption will be changed to: The slopes' **95%** confidence intervals are ± 0.007, 0.06 and 0.09 for MLS, HALOE and ACE-FTS, respectively.

**References**

Taylor, K. E: Summarizing multiple aspects of model performance in a single diagram, J. Geoph. Res., vol 106, no. D7, 7183-7192, doi:10.1029/2000JD900719, 2001.